# Sociodemographic and educational factors associated with mental health disorders in medical students of clinical years: A multicenter study in Peru

J. Pierre Zila-Velasque[1,2], Pamela Grados-Espinoza[1,2], Kateriny Margot Regalado-Rodríguez[3], Frank Sosa-Nuñez[4], Abimegireysch Alcarraz-Jaime[5], Andrea G. Cortez-Soto[6], Diego Chambergo-Michilot[7], Anderson N. Soriano-Moreno[8]*

1 Facultad de Medicina Humana, Universidad Nacional Daniel Alcides Carrión, Pasco, Peru, 2 Red Latinoamericana de Medicina en la Altitud e Investigación (REDLAMAI), Pasco, Peru, 3 Facultad de Medicina Humana, Universidad Nacional de Cajamarca, Cajamarca, Peru, 4 Escuela Profesional de Medicina Humana, Universidad Nacional de San Cristóbal de Huamanga, Ayacucho, Peru, 5 Escuela Profesional de Medicina Humana, Universidad Privada San Juan Bautista, Lima, Peru, 6 Facultad de Medicina Humana, Universidad Nacional San Luis Gonzaga, Ica, Peru, 7 Escuela de Medicina Humana, Facultad de Ciencias de la Salud, Universidad Científica del Sur, Lima, Peru, 8 Clinical and Epidemiological Research Unit, School of Medicine, Universidad Peruana Unión, Lima, Peru

* andsor19@gmail.com

## Abstract

### Objective

To identify sociodemographic and educational factors associated with mental health disorders in Peruvian medical students in clinical years.

### Methods

Cross-sectional study. We surveyed students from 24 Peruvian medical schools. We defined negative perception of educational environment as having a Dundee Ready Educational Environment Measure score below 100 points; we defined anxiety and depression as having more than 4 points on the Generalized Anxiety Disorder-7 and Patient Health Questionnaire-9 scales, respectively. Poisson regression with robust variance was used to assess the association between negative perception of educational environment and mental health alterations.

### Results

Among the 808 participants, the prevalence of anxiety and depression symptoms was 77.6% and 67.3%, respectively. Negative perception of the educational environment was 31.6%. The factors associated with anxiety were being male (PR = 0.95, 95% CI:0.91–0.98), previous medical condition (Prevalence ratios PR = 1.10, 95% CI:1.05–1.16), previous diagnosis of coronavirus disease 2019 (PR = 0.93, 95% CI: 0.93–0.94), being from highlands (PR = 1.11, 95% CI: 1.05–1.16), studying at a national university (PR = 0.90, 95% CI: 0.88–0.92), and negative perception of the educational environment (PR = 1.04, 95% CI:

information files. Database https://figshare.com/
articles/dataset/BD_-_Educational_Environment_
and_SM_Disorders_coding_xlsx/21874950
Commands used for statistical analysis https://
figshare.com/articles/dataset/DO_AE-SM_do/
21874965.

**Funding:** The author(s) received no specific
funding for this work.

**Competing interests:** The authors have declared
that no competing interests exist.

1.03–1.05), while factors associated with depression were being male (PR = 0.94, 95% CI:
0.93–0.95), previous medical condition (PR = 1.12, 95% CI: 1.08–1.17), type of university
(national) (PR = 0.95, 95% CI: 0.95–0.96), and negative perception of the educational envi-
ronment (PR = 1.11, 95% CI: 1.07–1.16).

## Conclusion

We found evidence that during the COVID-19 pandemic anxiety and depression are
prevalent among Peruvian medical students. Sociodemographic factors and negative
perception of educational environment were associated with the presence of these
conditions.

## Introduction

The pandemic caused by the coronavirus disease 19 (COVID-19) conditioned governments to
establish mandatory government public health policies; through social distancing and the sus-
pension of face-to-face educational activities [1]. Faced with an unforeseen situation, the conti-
nuity of higher education depended on the adaptation of the educational population to a
totally online or virtual system [2]. In Peru, the suspension of face-to-face activities in the edu-
cational field was ordered on March 12, 2020. Given this situation, the medical schools focused
on maintaining teaching through virtual channels and modifying the academic curriculum.
However, at the end of that same year, the number of infections was apparently lower and the
time was drawing near to return to face-to-face teaching. But the second wave of infections
came by chance and in a loud way from December 2020 until approximately the end of June
2021 [3].

The educational environment (EE) is defined as the set of physical structures and human
relationships in which an educational community develops [4]. To succeed in the teaching-
learning process, the EE must have appropriate characteristics for exchanging information and
experiences, where they must acquire new skills necessary for their professional performance
[5]. Likewise, the EE is considered a determinant of the behavior and development of the stu-
dents that serve to adapt to the educational demands; through, strategies that can lead to burn-
out, stress and in general affect mental health and with it negative repercussions on their
academic performance and making wrong decisions [6]. With the pandemic, the new virtual
environment of higher education challenged students with new academic, technological, and
psychological demands that have been seen and this has generated an increase in mental health
disorders (anxiety and depression) in students [7]. Some factors associated with the develop-
ment of these mental disorders in students are the presence of comorbidities [8], place of resi-
dence [9], gender [10], and use of harmful substances [11]. However, we have not found
studies that evaluate the association with EE, especially in a context as important as the pan-
demic and Latin America.

Therefore, the restrictions, the lack of alternative methods to the traditional learning meth-
ods known before the pandemic (face-to-face education in its entirety), and the lack of adapta-
tion of universities to virtuality [12]; have generated adaptive difficulties in students, who
behave as a population vulnerable to mental health disorders [13]. In this sense, the present
investigation aims to determine the socio-educational factors associated with mental health
disorders in Peruvian students of human medicine from 24 universities, where a wide variety
of factors not studied in other studies were found.

## Methods

### Study design

The present study was conducted during the months of January to March 2021, a period in which the second wave of infections began in the country and clinical rotations of medical students in hospitals were still restricted in Peru [3].

### Study population

The population consisted of students from the 3rd to 7th year (clinical years) from 24 different universities in Peru; from the three natural regions (coast, mountains, and jungle) that have a medical school [14,15]. The sample size was calculated with a prevalence of mental health disorders of 50.0%, a confidence level of 95%, a precision of 5%, a design effect factor of 1, and considering a infinite population which gave us a number of participants of 384, which was exceeded in our study (808 participants) [16]. The type of sampling was the snowball type, which happened when "one subject gives the researcher the name of another, who in turn gives the name of a third, and so on" [17].

We considered students who belonged to courses superior to the third year, regular students (those with more than 12 credits) and who accepted the informed consent. We excluded students who belonged to years below the third because they do not study in the clinical field according to the curriculum in our country. Also, because the instrument Dundee Ready Education Environment Measure (DREEM) that was used to evaluate the educational environment focuses on asking about characteristics of the clinical environment.

### Instrument and data collection

The online survey was divided into four sections: 1) Informed consent; 2) sociodemographic data, and educational data; 3) DREEM questionnaire that evaluates the EE perceived by the students; 4) 7-item Generalized Anxiety Disorder Scale (GAD-7) that evaluates symptoms of generalized anxiety disorder (GAD); and 9-item Patient Health Questionnaire (PHQ-9) that evaluates depressive symptoms. To collect the data, we created a survey on the Google Forms platform that was sent to the study collaborators and through them we proceeded to share the survey with all groups of students, in addition to sharing it on social networks (Facebook, WhatsApp, Telegram, Instagram) by posting on the class pages of medical schools. Consent was obtained from the participants at the beginning of the survey. Prior acceptance triggered the subsequent sections. Otherwise, the survey was terminated. Participants were asked to complete the survey only once and as honestly as possible, as we limited the survey to one response per participant, which ensured that no multiple responses were given with consequent overestimation of the results.

### Anxiety and depressive symptoms

The GAD-7 scale was used to assess anxiety symptoms. It consists of 7 items, with scores ranging from 0 (not at all) to 3 (almost every day), so that the total score ranges go from 0 to 21 and, in turn, can be classified into 4 severity groups: minimal (0–4), mild (5–9), moderate (10–14) and severe (14–20) [18]. The instrument has a sensitivity (89%) and specificity (82%) [19]. The GAD-7 was validated in Peru with good reliability (Cronbach's alpha = 0.89) [20]. We defined the presence of anxiety with a cut-off point > 4.

Depression was assessed using with the PHQ-9. The Spanish version shows a sensitivity of 92% and a specificity of 89% [21]. This instrument was validated in Peru and was found with an adequate internal consistency (Cronbach's alpha = 0.903). It consists of 9 items and each

item is scored according to a Likert scale that ranges from 0 (no day) to 3 (almost every day). The PHQ-9 scores reflect 5 categories of depressive disorder severity: none (0–4 points), mild (5–9 points), moderate (10–14 points), moderately severe (15–19 points), and severe (20–27 points) [22]. We defined the presence of depression with a cut-off point > 4.

### Educational factors

The perception of the EE was assessed with the DREEM instrument as this is considered the most optimal instrument for its evaluation. The instrument was validated and developed in Peru, with high reliability (internal consistency); Cronbach's alpha = 0.93 [23]. It involves 50 items. It is answered with a 5-level Likert-type scale: completely agree (4 points), agree (3 points), not sure or no opinion (2 points), disagree (1 point), and completely disagree (0 points). Negative perception of the EE was defined as a score between 0 and 50, with many problems between 51–100, more positive than negative between 101–150, and excellent between 151–200 [24,25]. Among the items of this instrument, we will find questions on the identification of areas of EE strengths and weaknesses, academic performance, self-perceived competence to work, aspects of the work environment, and others [26].

### Biostatistical methods

Categorical variables were described as frequencies and percentages, and continuous variables as mean with SD (standard deviation) or media with IQR (interquartile range) depending of the distribution of the data. We performed chi-square to determine the association between the sociodemographic and educational factors with anxiety and depression. In addition, simple and multiple regression models were created using generalized linear models (GLM) with Poisson distribution, robust variance, logarithmic link function and grouping by type of university to obtain prevalence ratios (PR) with 95% confidence intervals. The multivariate multiple model analysis was adjusted for all variables entered in the simple regression. Data analysis was carried out in STATA version 16.1 (College Station, TX: StataCorp LL).

### Ethical aspects

The study was approved by the institutional ethics committee of the Universidad Peruana Unión (Code: 2020-CEUPeU-00047). Informed consent was obtained from each participant and the data were anonymous and confidential. Only the researchers had and will have access to the database.

## Results

A total of 808 students were included. The median age was 23 (IQR: 21–25) years. More than half of them were women (56.4%) and came from cities located in the coast (58.0%). A total of 64.9% were studying at a private university. 15.4% reported having a history of the disease and 17.0% reported having or having had a diagnosis of COVID-19. More than half of the students reported that their motives for pursuing the degree were to be a person of integrity and useful in society (29.0%), to achieve a satisfying profession (27.0%), and to develop their skills (21.0%). In addition, 30.4% of students were in their fourth year and 11.2% were in internship. The prevalence of anxiety and depression symptoms was 77.6% and 67.3%, respectively. The prevalence of severe anxiety and depression symptoms was 13.9% and 5.9%, respectively (Table 1). The prevalence of good EE was 68.3%, and a large proportion (63.3%) had a more positive than negative perception of EE (Fig 1).

**Table 1. Socieducational variables of the medical students (n = 808).**

| Characteristics | N (%) |
|---|---|
| **Age (years)*** | 23 (21–25) |
| **Female sex** | 456 (56.4) |
| **Marital status** | |
| Single | 662 (81.9) |
| Engaged/Married/Other | 146 (18.1) |
| **Place of residence** | |
| House | 636 (78.7) |
| Other | 172 (21.3) |
| **Educational institution of origin** | |
| National/State | 326 (40.3) |
| Private/Private | 482 (59.7) |
| **Work while studying** | 156 (19.3) |
| **Previous medical condition** | 125 (15.4) |
| **Diagnosis of COVID-19** | 138 (17.0) |
| **Reasons for studying medicine** | |
| To develop my talents and skills | 436 (21.0) |
| To achieve a profession that satisfies me | 553 (27.0) |
| To be a person of integrity and useful to society. | 594 (29.0) |
| To please my parents | 128 (6.4) |
| To become independent from my family | 244 (12.1) |
| To achieve status and prestige | 295 (15.0) |
| Other | 13 (0.6) |
| **Region of origin** | |
| Coast | 469 (58.0) |
| Highlands | 289 (35.8) |
| Jungle | 50 (6.2) |
| **Studying with full academic load** | 643 (79.5) |
| **Current academic stage** | |
| 3rd year | 116 (14.3) |
| 4th year | 246 (30.5) |
| 5th year | 188 (23.2) |
| 6th year | 167 (20.7) |
| 7th year | 91 (11.3) |
| **Type of university** | |
| National or State | 283 (35.1) |
| Private or Private | 525 (64.9) |
| **Educational Environment (EE)**** | 110,7 (30) |
| Good | 552 (68.3) |
| Bad | 256 (31.7) |
| **Presence of anxiety** | 627 (77.6) |
| Mild | 297 (36.7) |
| Moderate | 217 (26.9) |
| Severe | 113 (14.0) |
| **Presence of Depression** | 544 (67.3) |
| Mild | 275 (34.0) |
| Moderate | 124 (15.3) |
| Moderately severe | 97 (12.0) |

(*Continued*)

**Table 1.** (Continued)

| Characteristics | N (%) |
|---|---|
| Severe | 48 (6.0) |

*Median—Interquartile range (IQR).

**Mean—Standard deviation.

In bivariate analysis, anxiety was associated with regarding gender, previous medical condition, COVID-19 diagnosis, region of origin, full academic load and type of university. Regarding depression, significant differences were found for gender, previous medical condition, and type of university, all according to bivariate analysis (Table 2).

In the multivariate analysis, we found that male students (PR: 0.95; 95% CI: 0.91–0.98), previous medical condition (PR: 1.10; 95% CI: 1.05–1.16), diagnosis of COVID-19 (PR: 0.93; 95% CI: 0.93–0.94), being from highlands (PR: 1.11; 95% CI: 1.05–1.16) and jungle regions (PR: 1.12; 95% CI: 1.09–1.15), taking a full academic load (PR: 1.03; 95% CI: 1.01–1.05), studying at a national or state university (PR: 0.90; 95% CI: 0.88–0.92) and presenting a bad educational environment (PR: 1.04; 95% CI: 1.03–1.05) were statistically associated anxiety (Table 3).

On the other hand, we found that male students (PR: 0.94; 95% CI: 0.93–0.95), previous medical condition (PR: 1.12; 95% CI: 1.08–1.17), studying at a national or state university (PR: 0.95; 95% CI: 0.95–0.96) and presenting a bad educational environment (PR: 1.11; 95% CI: 1.07–1.16) were associated with depression (Table 3).

## Discussion

This cross-sectional study evaluated the sociodemographic and educational factors associated with mental health disorders in 24 medical schools in Peru during the COVID-19 pandemic. We found high overall rates of anxiety symptoms (7 out of 10) and depression symptoms (6 out of 10). In addition, we found that being male, presenting a previous medical condition, having a diagnosis of COVID-19, coming from the highlands or jungle region, to take a full

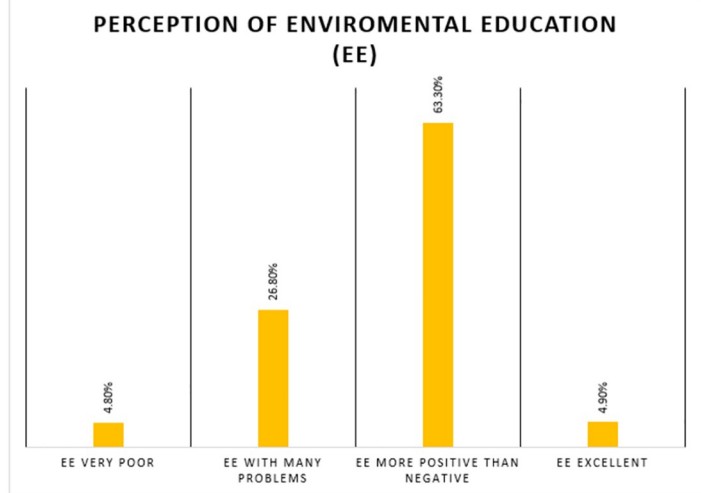

**Fig 1. Perception of environmental education.**

**Table 2. Socieducational factors associated with mental health disorders in medical students in Peru in a bivariate analysis.**

| Variables | Anxiety | | | Depression | | |
|---|---|---|---|---|---|---|
| | No | Yes | p-value | No | Yes | p-value |
| | n (%) | n (%) | | n (%) | n (%) | |
| **Gender*** | | | **0.002** | | | **0.001** |
| Female | 84 (18.4) | 372 (81.5) | | 127 (27.8) | 329 (72.1) | |
| Male | 97 (27.5) | 255 (72.4) | | 137 (38.9) | 215 (61.0) | |
| **Marital Status*** | | | 0.110 | | | 0.359 |
| Single | 141 (21.3) | 521 (78.7) | | 221 (33.3) | 441 (66.6) | |
| Engaged—Married | 40 (27.4) | 106 (72.6) | | 43 (29.4) | 103 (70.5) | |
| Varieties | 31 (37.3) | 52 (62.6) | | 37 (44.5) | 46 (55.4) | |
| **Place of residence*** | | | 0.753 | | | 0.687 |
| House | 144 (22.6) | 492 (77.3) | | 210 (33.0) | 426 (66.9) | |
| Other | 37 (21.5) | 135 (78.4) | | 54 (31.4) | 118 (68.6) | |
| **Previous medical condition** | | | **<0.001** | | | **<0.001** |
| No | 175 (25.6) | 508 (74.3) | | 247 (36.6) | 436 (63.8) | |
| Yes | 6 (4.8) | 119 (95.2) | | 17 (13.6) | 108 (86.4) | |
| **Diagnosis of COVID-19*** | | | **0.003** | | | 0.328 |
| No | 137 (20.4) | 533 (79.5) | | 214 (31.9) | 456 (68.0) | |
| Yes | 44 (31.8) | 94 (68.1) | | 50 (36.2) | 88 (63.7) | |
| **Region of origin*** | | | **0.004** | | | 0.378 |
| Coast | 123 (26.2) | 346 (73.7) | | 156 (33.2) | 313 (66.7) | |
| Highlands | 46 (15.9) | 243 (84.0) | | 88 (30.4) | 201 (69.5) | |
| Jungle | 12 (24.0) | 38 (76.0) | | 20 (40.0) | 30 (60.0) | |
| **Full academic load*** | | | **0.036** | | | 0.987 |
| No | 47 (28.4) | 118 (71.5) | | 54 (32.7) | 111 (67.2) | |
| Yes | 134 (20.8) | 509 (79.1) | | 210 (32.6) | 433 (67.3) | |
| **Working while studying*** | | | 0.089 | | | 0.995 |
| No | 154 (23.6) | 498 (76.3) | | 213 (32.6) | 439 (67.3) | |
| Yes | 27 (17.3) | 129 (82.6) | | 51 (32.6) | 105 (67.3) | |
| **Reasons for studying medicine*** | | | | | | |
| To develop my talents and skills | 57 (20.7) | 218 (79.2) | 0.412 | 85 (30.9) | 190 (69.0) | 0.442 |
| To achieve a profession that satisfies me | 52 (23.4) | 170 (76.5) | 0.668 | 73 (32.8) | 149 (67.1) | 0.938 |
| To be a person of integrity and useful to society. | 36 (24.6) | 110 (75.3) | 0.470 | 56 (38.3) | 90 (61.6) | 0.106 |
| To please my parents | 15 (11.7) | 113 (77.6) | **0.002** | 36 (28.1) | 92 (71.8) | 0.232 |
| To become independent from my family | 38 (20.7) | 145 (79.2) | 0.546 | 57 (31.6) | 126 (67.8) | 0.617 |
| Other | 2 (22.3) | 3 (77.6) | 0.341 | 1 (20.0) | 4 (80.0) | 0.546 |
| **Current academic stage*** | | | 0.324 | | | 0.060 |
| 3rd year | 18 (15.5) | 98 (84.4) | | 25 (21.5) | 91 (78.4) | |
| 4th year | 61 (24.8) | 185 (75.2) | | 85 (34.5) | 161 (65.4) | |
| 5th year | 46 (24.4) | 142 (75.5) | | 68 (36.1) | 120 (63.8) | |
| 6th year | 35 (20.9) | 132 (79.9) | | 52 (31.1) | 115 (68.8) | |
| 7th year | 21 (23.0) | 70 (76.9) | | 34 (37.3) | 57 (62.6) | |
| **Type of university*** | | | **<0.001** | | | **0.001** |
| Private | 43 (15.1) | 240 (84.8) | | 71 (25.0) | 212 (74.9) | |
| National or State | 138 (26.2) | 387 (73.7) | | 193 (36.7) | 332 (63.2) | |
| **Educational environment*** | | | 0.061 | | | **<0.001** |
| Good | 134 (24.2) | 418 (75.7) | | 216 (39.1) | 336 (60.8) | |
| Bad | 47 (18.3) | 209 (81.6) | | 48 (18.7) | 208 (81.2) | |

* p-value of categorical variables calculated with the Chi-Square test.

**Table 3. Socieducational factors associated with mental health disorders in Peruvian medical students in a multivariate analysis.**

| Variables | Anxiety | | | | | | Depression | | | | | |
|---|---|---|---|---|---|---|---|---|---|---|---|---|
| | Simple regression | | | Multiple regression | | | Simple regression | | | Multiple regression | | |
| | PR | IC 95% | p* value | PR | IC 95% | p* value | PR | IC 95% | p* value | PR | IC 95% | p* value |
| **Gender** | | | | | | | | | | | | |
| Female | Ref. | | | Ref. | | | Ref. | | | Ref. | | |
| Male | 0.95 | 0.92–0.97 | **<0.001** | 0.95 | 0.91–0.98 | **0.010** | 0.94 | 0.93–0.95 | **<0.001** | 0.94 | 0.93–0.95 | **<0.001** |
| **Previous medical condition** | | | | | | | | | | | | |
| No | Ref. | | | Ref. | | | Ref. | | | | | |
| Yes | 1.10 | 0.92–0.97 | **<0.001** | 1.10 | 1.05–1.16 | **<0.001** | 1.12 | 1.08–1.17 | **<0.001** | 1.12 | 1.08–1.17 | **<0.001** |
| **Diagnosis of COVID-19** | | | | | | | | | | | | |
| No | Ref. | | | Ref. | | | No entry to the model | | | No entry to the model | | |
| Yes | 0.93 | 0.93–0.94 | **<0.001** | 0.93 | 0.93–0.94 | **<0.001** | | | | | | |
| **Region of origin** | | | | | | | | | | | | |
| Coast | Ref. | | | Ref. | | | No entry to the model | | | No entry to the model | | |
| Highlands | 1.10 | 1.06–1.14 | **<0.001** | 1.11 | 1.05–1.16 | **<0.001** | | | | | | |
| Jungle | 1.11 | 1.08–1.15 | **<0.001** | 1.12 | 1.09–1.15 | **<0.001** | | | | | | |
| **Full academic load** | | | | | | | | | | | | |
| No | Ref. | | | Ref. | | | No entry to the model | | | No entry to the model | | |
| Yes | 1.03 | 1.00–1.05 | **0.006** | 1.03 | 1.01–1.05 | **0.001** | | | | | | |
| **Reasons for studying medicine** | | | | | | | | | | | | |
| To develop my talents and skills. | 1.00 | 0.99–1.01 | 0.513 | No entry to the model | | | No entry to the model | | | No entry to the model | | |
| To achieve a profession that satisfies me. | 0.99 | 0.95–1.03 | 0.942 | | | | | | | | | |
| To be a person of integrity and useful to society. | 0.97 | 0.94–1.00 | 0.093 | | | | | | | | | |
| To please my parents | 1.06 | 1.00–1.12 | **0.020** | | | | | | | | | |
| To become independent from my family. | 1.00 | 0.97–1.03 | 0.708 | | | | | | | | | |
| To achieve status and prestige. | 0.95 | 0.84–1.08 | 0.522 | | | | | | | | | |
| Other | 0.96 | 0.71–1.30 | 0.826 | | | | | | | | | |
| **Type of university*** | | | | | | | | | | | | |
| Private or Private | Ref. | | | Ref. | | | Ref. | | | Ref. | | |
| National or State | 0.90 | 0.89–0.92 | **<0.001** | 0.90 | 0.88–0.92 | **<0.001** | 0.95 | 0.95–0.96 | **<0.001** | 0.95 | 0.95–0.96 | **<0.001** |
| **Educational environment*** | | | | | | | | | | | | |
| Good | Ref. | | | Ref. | | | Ref. | | | Ref. | | |
| Bad | 1.04 | 1.02–1.05 | **<0.001** | 1.04 | 1.03–1.05 | **<0.001** | 1.11 | 1.07–1.16 | **<0.001** | 1.11 | 1.07–1.16 | **<0.001** |

*p-values obtained with Generalized Linear Models (GLM), fam Poisson, log link function, robust variance, PR: Prevalence ratio.

course load, studying at a national or state university, and presenting a negative perception of the educational environment was associated with anxiety; and being male, presenting a previous medical condition, studying at a national or state university, and presenting a bad educational environment were associated with depression.

## High prevalence of depression and anxiety

We found a higher prevalence of depression compared to other studies. In Shanghai universities, a prevalence of 46.2% was found, with males being the least affected; in Czech and Slovak university students, 52% and 47%, respectively [27–29]. However, in relation to the prevalence of mental health symptoms, our results are similar to those reported in a study carried out in Morocco that showed high levels of mental health symptoms especially women, who are in the preclinical stage and live in regions with a high prevalence of COVID-19 cases, where the

prevalence of anxiety and depression was 77.6% and 67.3%, respectively [30]. Similarly, another study reported an anxiety prevalence of 75.4% and a significant association with female sex, but this study was conducted on first-year university students [31]. However, they are contrary to a study that revealed a higher prevalence of anxiety and depression in face-to-face classes (42.3% and 49.3%, respectively) than in virtual classes (15.5% and 27.6%) [1]. A study in Peru during the pandemic that reported a prevalence of anxiety and depression of 47.6% and 47.3%, respectively, in young university students [32]. Denoting that in the pandemic context, the development of mental health symptoms was conditioned by the dissatisfaction associated with the maladjustment of universities to the virtual environment of education, which in our country occurred in 65.0% of students [12]. Another issue that could contribute to the high prevalence of mental health problems is the interruption of clinical rotations, which raises concerns about the potential impact on physicians' professional development [33].

## Sociodemographic factors associated with mental health disorders

Being male has presented a higher probability of presenting depression compared to women, a result similar to a study conducted in Asia [34], but, different from what was found in another where the most affected population were women [35], similarly, another study found no differences in relation to gender [36]. It has been reported that women are more likely to develop depressive symptoms [28,37] as opposed to men, because they have a lower threshold for developing these symptoms [38], however, our results add to the current literature regarding this association. Our result could be explained due to the fact that that men have been seen to have greater loneliness, greater economic distress and lower levels of resilience that predispose the development of these symptoms [39], a result that should be taken with caution because men with symptoms of health disorders patients do not seek treatment due to stigmatization, a situation that leads to the chronification of symptoms and the risk of committing suicide [40,41].

Having a history of illness is associated up to 10 and 12 times more with developing anxiety and depression, respectively. Similar to what was found in the Peruvian population, which was higher in women and young people [35], it is also supported by the result of another study carried out in Malaysian university students [42], showing that taking or living with health comorbidities predisposes to the condition mental [43].

People with a diagnosis of COVID-19 were 7 times more likely to present anxiety. This was different from a study of French university students who were 45 times more likely [44], a result similar to that evidenced in the Canadian population up to 55 times more [45] highlighting that this study was conducted in the general population. Our result could be explained by the fact that presenting a disease with high stress load such as COVID-19 conditions the increased production of cortisol as a result of a dysfunction of the hypothalamic-pituitary-adrenal axis, which keeps the organism in a constant development of anxiety in addition to other mechanisms that have not yet been elucidated [46].

Students who lived in regions such as the mountains and the jungle presented up to 11 times more likely to develop mental health symptoms, where 41.8% of students came from these regions characterized by high poverty (44.9% and 34.4%) of the Sierra and Selva, respectively [47], the same result evidenced at the educational level where approximately 25.0% of the residents present low pedagogical level [48], associated with situations that condition the development of mental health symptoms due to the little knowledge that has been able to be developed about COVID-19 [49,50], which has possibly generated greater uncertainty.

## Educational factors associated with mental health disorders

Students with a negative perception of the EE presented more anxiety and depression symptoms. University EE is known to have a significant impact on education. It is considered a determining factor in the academic development of the medical student. As well as the expression of the administrative management of the university and the quality of the study plan of the career [51]. Similarly, a study from Malaysia evaluated the association between EE and psychological distress found that a positive environment has direct influences on the psychological health of medical students [52]. Therefore, negative EE could lead to emotional instability or burnout resulting in lower academic performance [53], affecting their emotional stability and motivation to continue with their careers [54].

Our study found a lower score in the perception of the level of EE compared to a study carried out in Saudi Arabia, which obtained a mean score of 122.4, a result that could be due to the context prior to the pandemic [55]. In another study conducted in India, a total score greater than 130.6 was found, indicating a higher quality of the EE. This difference may be due to the pre-pandemic context, having a smaller sample and conducting the study in every academic year [56]. However, it should be mentioned that the DREEM instrument has questions about the clinical setting. Therefore, it cannot be applied to students entering medical school to take basic and general science courses. Our results are also lower than those of other Latin American medical schools. In Chile they found a result that ranged between 103.1 and 126.9 points. Similarly, a study conducted in Argentina found total scores for the 1st, 3rd, and 5th academic years ranging between 149.6 and 136.6 [57]. In Colombia, the total mean was 152.0, but it should be noted that this study was only conducted in one course [58]. This suggests that these countries have better curriculum plans than ours. However, it is worth mentioning that all the studies, including ours, presented a more positive than negative perception.

Students who came from a national university are 10 times more likely to develop some of the mental health symptoms. The national universities in our country correspond to more than 52.5% of the 40 faculties that exist [14]. National universities have fewer student health support programs and less connectivity to Internet networks [59] which leads to maladaptation to virtual education, a situation that has led to greater stress in university students in general [12] and even greater anxiety in medical students [2]. The medical schools of private universities, on the other hand, have better resources and were able to present a better adaptation to EE [16]. However, we have not found studies that evaluate this variable and have demonstrated this association, therefore, our results add to the current literature.

## Recommendations

The current context of COVID-19 affects the mental health of the student population; due to increased loneliness, mourning, and hopelessness. Also, there is concern about academic overload, ineffective communication with teachers, the future of the career, the delay in clinical rotations, and the fear of the infection spreading [30]. For this reason, it is imperative to improve EE in medical schools, since a constant and effective evaluation of online learning in universities is considered a potential protective factor for symptoms of anxiety and depression. During the pandemic, increased stress due to virtual classes, thoughts of abandonment, and decreased productivity were also perceived; these are considered potential risk factors for increased anxiety and depression [60]. For all of the above, it is recommended to improve EE through the implementation of curricular restructuring programs, mental health counseling, and well-being to reverse these consequences that affect academic performance.

It is suggested to carry out EE evaluations per academic year in the universities to estimate it's the perceptions of students and thus intensify psychological support in universities [61]. At

the same time, generate a culture of acceptance, concern and destigmatization of mental disorders in university students in crucial in every university [62]. Psychological counseling is indispensable, especially during the pandemic period or high-stress events, and measures must be taken to address mental health issues, such as providing remote counseling for students.

The experience shown in the study is valuable to be able to periodically monitor curricular changes, as well as to evaluate the pedagogical innovation that can be implemented. Annual studies should be conducted, by academic cycle, region of origin and type of university, focusing on factors associated with mental health symptoms.

## Limitations and strengths

Some of the limitations were the cross-sectional design of our study, which does not allow us to identify causal relationships between the study variables. The symptoms of anxiety and depression in this study were evidenced through the analysis of the universal self-assessment scale, which differed from the DSM-5 diagnostic criteria; therefore, the results of this study are for reference only and cannot be used as a guide for clinical diagnosis and treatment; however, it provides us with updated information on the burden of mental health in a particular population, medical students. Our study may present biases due to the lack of representativeness and the fact that the sampling was convenience sampling; however, we included students from more than half of the licensed medical schools (24 out of 40). Finally, our study may present measurement biases, since the questionnaires are self-administered, but as a strength, the instrument used to measure EE was applied to a larger sample compared to other published studies on the subject, and was applied to students in the clinical stage, therefore it can provide the information requested by the instrument.

On the other hand, to our knowledge, our study is the first study that evaluates the perception of EE and the prevalence of mental health disorders during the COVID-19 pandemic, and in a large and varied population, Peruvian medical students from 24 of the 40 medical schools in Peru, including the three regions of Peru, different by their economic status and quality of education, important in understanding the results. This study serves as a reference for implementing new educational environment strategies, and is further evidence that medical students deserve special attention to their mental health. This study also serves as a basis for further studies that wish to assess the mental health of students, in order to convince those in charge of education to improve the educational environment and reach conventional agreements with students to reverse the alarming numbers of mental illnesses they may have, which can alter the course of their lives and careers.

## Conclusion

We found evidence that during the COVID-19 pandemic the prevalence of anxiety and depression was elevated, and sociodemographic and educational factors were associated with the presence of these conditions. Educators and researchers may take in account these factors to improve mental health among medical students.

## Supporting information

**S1 Dataset. Database.**
(DOCX)

**S2 Dataset. Commands used for statistical analysis.**
(DOCX)

## Acknowledgments

We thank the NOBIOM group that is part of REDLAMAI for providing us with excellent collaborators from every university in the country.

## Author Contributions

**Conceptualization:** J. Pierre Zila-Velasque, Pamela Grados-Espinoza, Anderson N. Soriano-Moreno.

**Data curation:** J. Pierre Zila-Velasque, Pamela Grados-Espinoza.

**Formal analysis:** J. Pierre Zila-Velasque, Anderson N. Soriano-Moreno.

**Funding acquisition:** Anderson N. Soriano-Moreno.

**Investigation:** J. Pierre Zila-Velasque, Pamela Grados-Espinoza, Kateriny Margot Regalado-Rodríguez, Frank Sosa-Nuñez, Abimegireysch Alcarraz-Jaime, Andrea G. Cortez-Soto, Diego Chambergo-Michilot.

**Methodology:** J. Pierre Zila-Velasque, Pamela Grados-Espinoza.

**Project administration:** J. Pierre Zila-Velasque, Pamela Grados-Espinoza, Anderson N. Soriano-Moreno.

**Resources:** Anderson N. Soriano-Moreno.

**Software:** J. Pierre Zila-Velasque.

**Supervision:** Anderson N. Soriano-Moreno.

**Validation:** J. Pierre Zila-Velasque, Kateriny Margot Regalado-Rodríguez, Frank Sosa-Nuñez, Abimegireysch Alcarraz-Jaime, Andrea G. Cortez-Soto, Diego Chambergo-Michilot.

**Visualization:** Kateriny Margot Regalado-Rodríguez, Frank Sosa-Nuñez, Abimegireysch Alcarraz-Jaime, Andrea G. Cortez-Soto, Diego Chambergo-Michilot.

**Writing – original draft:** J. Pierre Zila-Velasque, Pamela Grados-Espinoza, Kateriny Margot Regalado-Rodríguez, Frank Sosa-Nuñez, Abimegireysch Alcarraz-Jaime, Andrea G. Cortez-Soto, Diego Chambergo-Michilot.

**Writing – review & editing:** J. Pierre Zila-Velasque, Pamela Grados-Espinoza, Kateriny Margot Regalado-Rodríguez, Frank Sosa-Nuñez, Abimegireysch Alcarraz-Jaime, Andrea G. Cortez-Soto, Diego Chambergo-Michilot, Anderson N. Soriano-Moreno.

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
