## [Decision Letter · Decision Letter 0]

28 Nov 2022

PONE-D-22-28915Educational environment and mental health in medical students: a multicenter study conducted in 24 Peruvian universitiesPLOS ONE

Dear Dr. Soriano,

Thank you for submitting your manuscript to PLOS ONE. After careful consideration, we feel that it has merit but does not fully meet PLOS ONE’s publication criteria as it currently stands. Therefore, we invite you to submit a revised version of the manuscript that addresses the points raised during the review process.

Please submit your revised manuscript by Jan 12 2023 11:59PM.  If you will need more time than this to complete your revisions, please reply to this message or contact the journal office at plosone@plos.org. Please include the following items when submitting your revised manuscript:A rebuttal letter that responds to each point raised by the academic editor and reviewer(s). You should upload this letter as a separate file labeled 'Response to Reviewers'.A marked-up copy of your manuscript that highlights changes made to the original version. You should upload this as a separate file labeled 'Revised Manuscript with Track Changes'.An unmarked version of your revised paper without tracked changes. You should upload this as a separate file labeled 'Manuscript'.

We look forward to receiving your revised manuscript.

Kind regards,

Stephan Doering, M.D.

Academic Editor

**Journal Requirements: **

Reviewers' comments:

Reviewer's Responses to Questions

**Comments to the Author**

1. Is the manuscript technically sound, and do the data support the conclusions?

Reviewer #1: Yes

Reviewer #2: Yes

2. Has the statistical analysis been performed appropriately and rigorously? 

Reviewer #1: I Don't Know

Reviewer #2: I Don't Know

3. Have the authors made all data underlying the findings in their manuscript fully available?

Reviewer #1: No

Reviewer #2: No

4. Is the manuscript presented in an intelligible fashion and written in standard English?

Reviewer #1: No

Reviewer #2: Yes

5. Review Comments to the Author

Reviewer #1: Thank you for inviting me to review this article. It is well written and has a high potential for publication. However, please find my comments below to improve the manuscript.

Title –

- It is ok but the word ‘multicenter’ and ’24 Peruvian universities’ might be repetitive. Therefore, the methodology of the study might replace and be additionally mentioned in the title.

Abstract

- Abbreviations are suggested to be clarified.

- The reported statistics should follow the journal’s format.

- The last sentence of the result might be considered as the discussion. Meanwhile, the conclusion could be shorter and clearly sum up the finding, discussion, and further suggestion.

Introduction

- The sequence of each paragraph might be rearranged to make the section easier for readers. For example, mental morbidities might be stated following the importance of EE which is directedly affected by the pandemic as the authors firstly mentioned.

- After that, mental illness among medical students should be focused. There are several previous studies in respect to such study population, including

(1) Zeng W, Chen R, Wang X, et al. Prevalence of mental health problems among medical students in China: A meta-analysis. Medicine (Baltimore). 2019;98(18):e15337. doi:10.1097/MD.0000000000015337

(2) Chiddaycha M, Wainipitapong S. Mental health among Thai medical students: Preadmission evaluation and service utilization. Health Sci Rep. 2021 Oct 9;4(4):e416. doi: 10.1002/hsr2.416. PMID: 34646944; PMCID: PMC8501376.

(3) Pham T, Bui L, Nguyen A, et al. The prevalence of depression and associated risk factors among medical students: An untold story in Vietnam. PLoS One. 2019;14(8):e0221432. doi:10.1371/journal.pone.0221432

(4) Puthran R, Zhang MWB, Tam WW, et al. Prevalence of depression amongst medical students: a meta-analysis. Med Educ. 2016;50(4):456-468. doi:10.1111/medu.12962

(5) Pacheco JP, Giacomin HT, Tam WW, Ribeiro TB, Arab C, Bezerra IM, Pinasco GC. Mental health problems among medical students in Brazil: a systematic review and meta-analysis. Braz J Psychiatry. 2017 Oct-Dec;39(4):369-378. doi: 10.1590/1516-4446-2017-2223. Epub 2017 Aug 31. PMID: 28876408; PMCID: PMC7111407.

. . and so on.

- Moreover, consequences of those mental conditions toward medical students could be stated (i.e., dropout - doi: 10.1186/s12909-022-03527-z / suicide - doi: 10.1097/ACM.0000000000002507 / and many more) to emphasize and justify your study of linkage between mental health and EE. Finally, the last paragraph then acclaimed your study justification and the objective of the study.

- Though I am not a native English user, I believe that the manuscript will gain benefit from English editing.

Methods –

- The authors should clarify whether they select clinical year students as a sample, but the title referred to any medical students.

- Page 4 Line 108-110 – Is it a typo?

- Page 5 Line 111-119 – The COVID-19 situation affecting medical education in Peru should be stated in the introduction instead of the method section.

- Reference of sample size calculation is recommended to be added.

- The snowball sampling is quite confusing. Please kindly explain more about this, because authors later stated about advertising the surveys through social network platforms.

- Inclusion and exclusion criteria should be clearly stated. Also, the timeline of the study should be mentioned for the surveillance timepoint is vital for mental health screening according to the nature of academic/examination effect throughout the year, and the data of each study year would be a better representative if they were collected at the same timepoint.

- I think some details are redundant (i.e., the process in the Google Forms) and could be shorten or removed.

- Some items measured by each questionnaire should be exemplified for readers might easily view the scopes of each test, especially the EE which might not be familiar in readers from the field of psychiatry.

- What is ES (Page 7 Line 179)?

- Statistical analysis subheading should provide general statistical usage of the manuscript and not link with the finding. I am not sure about the median and IQR used in terms of age because it is not normally-distributed. Thus, I recommend more general statement such as ‘either Mean +/- SD or Median (IQR) was used regarding the distribution of data, normally and not-normally distributed, respectively’.

- Please kindly check terms ‘Multivariate’ and ‘Multiple’ analysis. Also, I am not sure whether authors wanted to infer ‘Bivariate’ in Page 7 Line 189, or not.

Results –

- Please kindly check the reported age, is it a mean with IQR?

- What is the difference between Half of them (56.4%) and More than half (58.0%) in Page 8 Line 204-205?

- Please kindly recheck the spelling (and meaning) in the table. For example, ‘Caracteristicas’ or ‘Typo of property’.

- Also, please kindly recheck the reported percent. For example in the marital status, if single is accountable for 81.9%, the reset should be 146/808 x 100 = 18.06 >> 18.1).

- Some definitions were unclear and needed explanation (i.e., human medicine or full academic load).

- Authors stated about bivariate analysis in Table 2 (Page 10 Line 221-222) which was not consistent with the declaration below the table mentioning about the chi-square test. If the bivariate analysis was done, I suggest to add an additional table with proper statistical details including OR, 95% CI, or P-value. Also, some of the variables might need the Fischer exact test. Please kindly reconfirm with statistical experts with all statistical analysis.

- I am not a statistician and uncertain about the term ‘Simple and multiple regression’ in multivariate analysis. Thus, I cannot give many comments about this issue. Also, please recheck the term used with the abstract either.

- Was P-value of being male and anxiety significant with the P-value 0.010?

- The Figure is not necessary. However, authors might elaborate some comparative variables including those with and without mental morbidities in the figure.

Discussion –

- The authors acclaimed that the finding was superior to previous studies (Page 15 Line 237). Please kindly provide evidence for the statement.

- The citation number 14 cannot be used to report more frequent depression in male students.

- Specifically, authors are recommended to discuss in the context of the clinical medical students during the COVID-19, which is unique and worth discussing.

- More strengths of this study should be mentioned.

- Limitations should include the generalizability for either sociocultural contexts, pandemic situations, or the clinical year only that recruited in the study.

Reference –

Please recheck the format; for example number 42,

Yt N, T EM, S W, D J, Cm W, Ha H. The association between COVID-19

546 diagnosis or having symptoms and anxiety among Canadians: A repeated cross547

sectional study. Anxiety, stress, and coping. 2021;34(5).

548 doi:10.1080/10615806.2021.1932837

Reviewer #2: Really interesting article, congrats to all authors. You were very honest and clear in describing the limitations and strengths of the study. Please, especially review grammar punctuation.

Some doubts and concerns are:

1- For adequate transparency, authors must make the data completely available except where there are legal and ethical concerns. The authors did not explain the reason for not sharing their anonymized data, you just denied it without justification. This is my major concern.

2- "There have been other studies that evaluated mental health worldwide, however, these studies have some limitations such as small size relevant because it has been shown that an insufficient sample leads to the estimation of a parameter with lower precision that generates wrong conclusions because they behave as the core of mental health assessment in any population group and are included in primary health care, not including other mental health variables (anxiety), not using validated questionnaires because the use of this type of instrument leads to false results that cannot be generalized, only including population from one university condition that does not lead to extrapolation of the results, and not having evaluated variables such as region of origin, type of university attended, which has been shown that in our country there is a large gap between the locations of each university and its type (national or particular) and EE which behaves as an influential variable in the professional development of the student, variables that are evaluated and analyzed in our study." - Very long sentence. This impairs its understanding.

3- Lines 108 to 110 are in Spanish: "sin embargo, esta 109 dispersion of faculties conditions that not all provide an education with 110 quality standards [20]."

4- Line 118: "the time in which the (DELETE THE) they kept the restriction"

5- "The sample size was calculated with a prevalence of mental health disorders of 50.0%" - Why did you consider this prevalence?

6- "We were unable to estimate a total population of human medicine students because this information is not available" - Can't you figure out how many medical schools there are in Peru, and how many vacancies are there in each of them? With this calculation you can estimate the total number of students.

7- How did you manage to ensure that all students from the schools included in the study received an invitation to participate in the research? Did you check name by name in whatsapp groups? Did you send it to an email bank?

8- Both the GAD-7 and PHQ-9 are widely adopted tools for screening for mental disorders, not diagnosis. Therefore, prefer to use "anxiety symptoms" and "depressive symptoms".

9 - Line 209: "región of origin" - spanish

10 - Line 209: "acaddemic load"  academic load

11- Lines 100 to 101: "where a wide variety of factors not studied in other studies were found." - Which factors?

12- Lines 67 to 68: "however, we have not found a study that evaluates the association with the educational environment (EE)" - https://doi.org/10.1186/s12909-022-03249-2 / https://doi.org/10.21203/rs.3.rs-2256756/v1

6. PLOS authors have the option to publish the peer review history of their article (what does this mean?). If published, this will include your full peer review and any attached files.

Reviewer #1: No

Reviewer #2: No

---

## [Author Response · Author response to Decision Letter 0]

26 Apr 2023

Dear Editor

We send you a cordial greeting from the research team.

Please find enclosed a response letter containing the answers to each of the recommendations/observations made by the reviewer, as well as the corrected and revised manuscript, in this last point we thought it appropriate to send the manuscript with the change control so that you can review each modified point. Finally, we sent a manuscript that is without the change control, which would be the final version. 

Response to recommendations: 

The paragraphs without highlighting correspond to the recommendations/comments made, and those highlighted to the responses with the modifications made. 

Review 1

Thank you for inviting me to review this article. It is well written and has a high potential for publication. However, please find my comments below to improve the manuscript.

Title – 

- It is ok but the word ‘multicenter’ and ’24 Peruvian universities’ might be repetitive. Therefore, the methodology of the study might replace and be additionally mentioned in the title.

We agree about the suggestion, you will see the change in the manuscript.

Abstract

- Abbreviations are suggested to be clarified.

We agree about the suggestion, you will see the change in the manuscript.

- The reported statistics should follow the journal’s format.

We agree about the observation, you will see the change in the manuscript.

- The last sentence of the result might be considered as the discussion.

We agree about the suggestion, you will see the change in the manuscript.

Meanwhile, the conclusion could be shorter and clearly sum up the finding, discussion, and further suggestion.

We agree about the suggestion, you will see the change in the manuscript.

Introduction

- The sequence of each paragraph might be rearranged to make the section easier for readers. For example, mental morbidities might be stated following the importance of EE which is directedly affected by the pandemic as the authors firstly mentioned.

- After that, mental illness among medical students should be focused. There are several previous studies in respect to such study population, including

(1) Zeng W, Chen R, Wang X, et al. Prevalence of mental health problems among medical students in China: A meta-analysis. Medicine (Baltimore). 2019;98(18):e15337. doi:10.1097/MD.0000000000015337

(2) Chiddaycha M, Wainipitapong S. Mental health among Thai medical students: Preadmission evaluation and service utilization. Health Sci Rep. 2021 Oct 9;4(4):e416. doi: 10.1002/hsr2.416. PMID: 34646944; PMCID: PMC8501376.

(3) Pham T, Bui L, Nguyen A, et al. The prevalence of depression and associated risk factors among medical students: An untold story in Vietnam. PLoS One. 2019;14(8):e0221432. doi:10.1371/journal.pone.0221432

(4) Puthran R, Zhang MWB, Tam WW, et al. Prevalence of depression amongst medical students: a meta-analysis. Med Educ. 2016;50(4):456-468. doi:10.1111/medu.12962

(5) Pacheco JP, Giacomin HT, Tam WW, Ribeiro TB, Arab C, Bezerra IM, Pinasco GC. Mental health problems among medical students in Brazil: a systematic review and meta-analysis. Braz J Psychiatry. 2017 Oct-Dec;39(4):369-378. doi: 10.1590/1516-4446-2017-2223. Epub 2017 Aug 31. PMID: 28876408; PMCID: PMC7111407.

. . and so on.

- Moreover, consequences of those mental conditions toward medical students could be stated (i.e., dropout - doi: 10.1186/s12909-022-03527-z / suicide - doi: 10.1097/ACM.0000000000002507 / and many more) to emphasize and justify your study of linkage between mental health and EE. Finally, the last paragraph then acclaimed your study justification and the objective of the study.

- Though I am not a native English user, I believe that the manuscript will gain benefit from English editing.

We agree about the suggestion, you will see the change in the manuscript. 

Methods 

- The authors should clarify whether they select clinical year students as a sample, but the title referred to any medical students.

We mention in this section that our population was students from the clinical years (3rd to 7th year) of 24 different universities in Peru, likewise, we have added this specific population in the title.

- Page 4 Line 108-110 – Is it a typo?

We apologize for this typographical error, the correct translation has been added to the manuscript, you will see the changes in the manuscript.

- Page 5 Line 111-119 – The COVID-19 situation affecting medical education in Peru should be stated in the introduction instead of the method section.

We agree with the suggestion and have moved the paragraph to the end of the introduction.

- Reference of sample size calculation is recommended to be added.

At this point, we do not agree with the suggestion because the size of the sample calculation was made by ourselves, not based on any specific bibliography.

- The snowball sampling is quite confusing. Please kindly explain more about this, because authors later stated about advertising the surveys through social network platforms.

We agree with the suggestion, so we have added the definition in this regard and its reference. You will see the changes in the manuscript.

- Inclusion and exclusion criteria should be clearly stated. Also, the timeline of the study should be mentioned for the surveillance timepoint is vital for mental health screening according to the nature of academic/examination effect throughout the year, and the data of each study year would be a better representative if they were collected at the same timepoint.

We agree about the suggestion, you will see the change in the manuscript. 

- I think some details are redundant (i.e., the process in the Google Forms) and could be shorten or removed.

At this point we feel it is important to mention this, because it shows how we collected the data for the study.

- Some items measured by each questionnaire should be exemplified for readers might easily view the scopes of each test, especially the EE which might not be familiar in readers from the field of psychiatry.

We agree about the suggestion, you will see the change in the manuscript. 

- What is ES (Page 7 Line 179)?

The correct is EE (educational environment), we apologize for the error. We have corrected it; you will see the change in the manuscript.

- Statistical analysis subheading should provide general statistical usage of the manuscript and not link with the finding. I am not sure about the median and IQR used in terms of age because it is not normally-distributed. Thus, I recommend more general statement such as ‘either Mean +/- SD or Median (IQR) was used regarding the distribution of data, normally and not-normally distributed, respectively’.

We thank you for your suggestion, we have taken it into account in the manuscript, you will see the changes.

- Please kindly check terms ‘Multivariate’ and ‘Multiple’ analysis. Also, I am not sure whether authors wanted to infer ‘Bivariate’ in Page 7 Line 189, or not.

We agree with your observation, the correct term is Bivariate Analysis, we have corrected it in the manuscript, and we have considered the term Multivariate as the best option. You will see the changes.

Results –

- Please kindly check the reported age, is it a mean with IQR?

It is the median with IQR. We have corrected this in the manuscript.

- What is the difference between Half of them (56.4%) and More than half (58.0%) in Page 8 Line 204-205? 

Thanks for the observation, the meanings are the same, therefore, we consider keeping "more than half". The percentages correspond according to each variable evaluated. ; You will see the change in the manuscript.

- Please kindly recheck the spelling (and meaning) in the table. For example, ‘Caracteristicas’ or ‘Typo of property’.

Thanks for the observation. It has been corrected the correct form about the variables. You will see the change in the manuscript.

- Also, please kindly recheck the reported percent. For example in the marital status, if single is accountable for 81.9%, the reset should be 146/808 x 100 = 18.06 >> 18.1).

Thank you for your comment. We have corrected the percentage values in each variable. You will see the change in the manuscript.

- Some definitions were unclear and needed explanation (i.e., human medicine or full academic load).

At this point we have added in the part of biostatistical methods the definition on the complete academic load, in relation to the variable human medicine we have decided to eliminate the word human because it is understood in the context of the population. You will see the change in the manuscript.

- Authors stated about bivariate analysis in Table 2 (Page 10 Line 221-222) which was not consistent with the declaration below the table mentioning about the chi-square test. If the bivariate analysis was done, I suggest to add an additional table with proper statistical details including OR, 95% CI, or P-value. Also, some of the variables might need the Fischer exact test. Please kindly reconfirm with statistical experts with all statistical analysis.

We thank you for your suggestion, however, we mention that Table 2 does show the P-value of each variable, and we performed a multivariate analysis which is shown in Table 3. We report the prevalence ratio instead of the odds ratio.

- I am not a statistician and uncertain about the term ‘Simple and multiple regression’ in multivariate analysis. Thus, I cannot give many comments about this issue. Also, please recheck the term used with the abstract either.

We appreciate your comment, however, the terms used (Simple and multiple) are widely used in multivariate statistical analysis, so we decided to keep them in the manuscript.

- Was P-value of being male and anxiety significant with the P-value 0.010?

At this point, we have only considered the results above 0.05. incase of being male, we emphasize that this association was diluted in the multiple regression.

- The Figure is not necessary. However, authors might elaborate some comparative variables including those with and without mental morbidities in the figure.

We agree with the suggestion, we decided to eliminate the figure.

Discussion –

- The authors acclaimed that the finding was superior to previous studies (Page 15 Line 237). Please kindly provide evidence for the statement.

We have added studies showing a low prevalence compared to ours. You will see the change in the manuscript.

- The citation number 14 cannot be used to report more frequent depression in male students. 

We have not used the reference as study to compare our result, It has only used as a study that had some limitation like we mentioned in the introduction by not using validated questionnaires. 

- Specifically, authors are recommended to discuss in the context of the clinical medical students during the COVID-19, which is unique and worth discussing.

We agree with the suggestion, we have focused in COVID-19 context. You will see the change in the manuscript.

- More strengths of this study should be mentioned.

We agree with the suggestion, we have added more strengths. You will see the change in the manuscript.

- Limitations should include the generalizability for either sociocultural contexts, pandemic situations, or the clinical year only that recruited in the study.

- We agree with the suggestion, we have added to the manuscript. You will see the change in the manuscript.

Reference –

 Please recheck the format; for example number 42,

Yt N, T EM, S W, D J, Cm W, Ha H. The association between COVID-19 diagnosis or having symptoms and anxiety among Canadians: A repeated cross-sectional study. Anxiety, stress, and coping. 2021;34(5). doi:10.1080/10615806.2021.1932837

We thanks your observation, we have checked all the references. 

Review 2

Really interesting article, congrats to all authors. You were very honest and clear in describing the limitations and strengths of the study. Please, especially review grammar punctuation.

Some doubts and concerns are:

1- For adequate transparency, authors must make the data completely available except where there are legal and ethical concerns. The authors did not explain the reason for not sharing their anonymized data, you just denied it without justification. This is my major concern.

At this point, we have shared our data in a public repository, which you can find under Supporting Information.

2- "There have been other studies that evaluated mental health worldwide, however, these studies have some limitations such as small size relevant because it has been shown that an insufficient sample leads to the estimation of a parameter with lower precision that generates wrong conclusions because they behave as the core of mental health assessment in any population group and are included in primary health care, not including other mental health variables (anxiety), not using validated questionnaires because the use of this type of instrument leads to false results that cannot be generalized, only including population from one university condition that does not lead to extrapolation of the results, and not having evaluated variables such as region of origin, type of university attended, which has been shown that in our country there is a large gap between the locations of each university and its type (national or particular) and EE which behaves as an influential variable in the professional development of the student, variables that are evaluated and analyzed in our study." - Very long sentence. This impairs its understanding.

We have decided to delete this paragraph, thank you for your suggestion.

3- Lines 108 to 110 are in Spanish: "sin embargo, esta 109 dispersion of faculties conditions that not all provide an education with 110 quality standards [20]."

That part it has been correct, you will see the changes in the manuscript.

4- Line 118: "the time in which the (DELETE THE) they kept the restriction"

We have done different modifications in the manuscript, you will see the changes.

5- "The sample size was calculated with a prevalence of mental health disorders of 50.0%" - Why did you consider this prevalence?

We have considered this prevalence because at the time we decided to do this study, we did not find a study that reported this prevalence in relation to our context, in addition to being used as a value when a real prevalence is not known.

6- "We were unable to estimate a total population of human medicine students because this information is not available" - Can't you figure out how many medical schools there are in Peru, and how many vacancies are there in each of them? With this calculation you can estimate the total number of students.

In the study we mentioned the number of medical schools, but the number of students per school is not provided publicly, even less in the private universities, which is why we could not estimate the total population.

7- How did you manage to ensure that all students from the schools included in the study received an invitation to participate in the research? Did you check name by name in whatsapp groups? Did you send it to an email bank?

In relation to this point, when conducting the study we launched a call for collaborators or students from each university, through meetings they showed us that they shared the link in each of their communication groups.

8- Both the GAD-7 and PHQ-9 are widely adopted tools for screening for mental disorders, not diagnosis. Therefore, prefer to use "anxiety symptoms" and "depressive symptoms".

We agree with the suggestion, we have added the word “symptoms”, you will see the changes in the manuscript.

9 - Line 209: "región of origin" – Spanish

We have checked the words; it has been translated into English.

10 - Line 209: "acaddemic load"  academic load

We have checked the words; it has been corrected.

11- Lines 100 to 101: "where a wide variety of factors not studied in other studies were found." - Which factors?

We refer to the variables found in the study such as male sex, history of disease, diagnosis of COVID-19, being from highland and jungle regions, having a full academic load and the type of university.

12- Lines 67 to 68: "however, we have not found a study that evaluates the association with the educational environment (EE)" - https://doi.org/10.1186/s12909-022-03249-2 / https://doi.org/10.21203/rs.3.rs-2256756/v1

 At this point, at the time of searching for studies we did not find them, but as there is evidence that they are, we decided to add the words " in our context as the Latin American".

---

## [Editor Report · Decision Letter 1]

15 May 2023

Sociodemographic and educational factors associated with mental health disorders in medical students of clinical years: a multicenter study in Peru

PONE-D-22-28915R1

Dear Dr. Soriano,

We’re pleased to inform you that your manuscript has been judged scientifically suitable for publication and will be formally accepted for publication once it meets all outstanding technical requirements.

Kind regards,

Stephan Doering, M.D.

Academic Editor

PLOS ONE
---

## [Editor Report · Acceptance letter]

16 Jun 2023

PONE-D-22-28915R1 

Sociodemographic and educational factors associated with mental health disorders in medical students of clinical years: a multicenter study in Peru 

Dear Dr. Soriano:

I'm pleased to inform you that your manuscript has been deemed suitable for publication in PLOS ONE. Congratulations! Your manuscript is now with our production department. 

Kind regards, 

on behalf of

Professor Stephan Doering 

Academic Editor

PLOS ONE